# Forest defoliator outbreaks alter nutrient cycling in northern waters

Samuel G. Woodman [1✉], Sacha Khoury[2], Ronald E. Fournier [3], Erik J. S. Emilson [3], John M. Gunn[4], James A. Rusak [5] & Andrew J. Tanentzap [1]

Insect defoliators alter biogeochemical cycles from land into receiving waters by consuming terrestrial biomass and releasing biolabile frass. Here, we related insect outbreaks to water chemistry across 12 boreal lake catchments over 32-years. We report, on average, 27% lower dissolved organic carbon (DOC) and 112% higher dissolved inorganic nitrogen (DIN) concentrations in lake waters when defoliators covered entire catchments and reduced leaf area. DOC reductions reached 32% when deciduous stands dominated. Within-year changes in DOC from insect outbreaks exceeded 86% of between-year trends across a larger dataset of 266 boreal and north temperate lakes from 1990 to 2016. Similarly, within-year increases in DIN from insect outbreaks exceeded local, between-year changes in DIN by 12-times, on average. As insect defoliator outbreaks occur at least every 5 years across a wider 439,661 $km^2$ boreal ecozone of Ontario, we suggest they are an underappreciated driver of biogeochemical cycles in forest catchments of this region.

[1] Ecosystems and Global Change Group, Department of Plant Sciences, University of Cambridge, Cambridge, UK CB2 3EA. [2] Forest Ecology and Conservation Group, University of Cambridge Conservation Research Institute, University of Cambridge, Cambridge, UK CB2 3QZ. [3] Natural Resources Canada, Canadian Forest Service, Great Lakes Forestry Centre, 1219 Queen St. E, Sault Ste. Marie, ON, Canada P6A 2E5. [4] Cooperative Freshwater Ecology Unit, Vale Living with Lakes Centre, Laurentian University, 935 Ramsey Lake Road, Sudbury, ON, Canada P3E 2C6. [5] Dorset Environmental Science Centre, Ontario Ministry of the Environment, Conservation and Parks, 1026 Bellwood Acres Road, Dorset, ON, Canada P0A 1E0. ✉email: sgw35@cam.ac.uk

Freshwaters are the main conduit for transporting macronutrients, including carbon (C) and nitrogen (N), between their major reservoirs on land and the oceans[1,2]. Thus, the nutrient status and subsequent functioning of freshwaters closely reflect their surrounding catchments[3]. C and N mostly enter freshwaters as dissolved fractions after the leaching of dead plant organic matter (OM) from soils and fresh foliar litter[4]. In lakes, dissolved organic carbon (DOC) regulates ecosystem structure and function[5], by affecting light penetration[6], thermal stability[7], contaminant toxicity[8], and food web nutrition[3]. Dissolved inorganic nitrogen (DIN; nitrates and ammonium) also influences lake ecosystem structure and function since it is a limiting nutrient for the growth of plants, algae, and bacteria[9], and thus available energy at the base of aquatic food webs[10]. Despite the importance of C and N fluxes, how they vary in lakes because of biotic disturbances in their surrounding catchments remains poorly understood. Previous work has focused on wood-boring insects, like bark beetles, that kill large numbers of trees and cause dead OM to accumulate on the landscape[11]. These events can subsequently release large pulses of C and N into downstream waters owing to both reduced N uptake[12–15] and increased litter inputs[16] from dead trees into some downstream waters[12,17]. By contrast, defoliator insects consume foliage without killing trees, and their consequences for nutrient cycling from land into water remain unknown.

Outbreaks of insect defoliators cause high severity, periodic disturbances that can alter the quantity[18], quality[19], and timing[20] of C and N inputs into lakes from surrounding catchments. In years without outbreaks, C and N entering lakes often come from decaying leaf/needle litter and typically peak in quantity during autumn[21]. Insect defoliation can reduce the amount of foliar litter C and N available for export to lakes through the ingestion of leaves/needles[18,22]. For example, widespread outbreaks of invasive gypsy moths (*Lymantria dispar dispar*) and forest tent caterpillars (*Malacasoma disstria*) can completely defoliate temperate forest canopies during early summer[23] when their feeding peaks[24]. Defoliators can, however, offset lower N inputs during autumn leaf senescence by releasing frass onto soils as a by-product of feeding. Frass is N-rich because insect defoliators are inefficient at assimilating foliar N[25]. N is also readily available in foliage during insect feeding that occurs before trees resorb foliar N to reduce losses during leaf senescence[25]. For example, frass produced from oak leaves (*Quercus rubra*) has lower C:N ratio than the corresponding leaf litter of 20:1 vs 24:1, respectively, and so is potentially more biolabile[26,27]. Unlike C and N, there may be little change in phosphorus (P) cycling from defoliation as it relies more on atmospheric inputs, wetland flowpaths, and internal recycling than plant litter fluxes[28]. N:P ratios may therefore shift downstream, with consequences for lake food webs as P limitation becomes more prevalent[29]. Foliar chemistry, more generally, also varies among tree species[30], so, the species composition of defoliated catchments is likely an overriding factor on the biolability and biogeochemical fate of both litterfall and resulting frass[31].

Here, we report lower DOC and higher DIN concentrations in northern lake waters during outbreaks of insect defoliators in surrounding catchments, particularly when containing higher proportions of deciduous stands. We analyzed 32-years of insect outbreak surveys and monthly lake water chemistry data from 12 single-lake catchments (18–1045 ha in size) across Ontario, Canada. Using mixed-effects models, we tested the effects of defoliating insects by explicitly accounting for temporal autocorrelation in water chemistry and variation among sites because of disturbance history and landscape characteristics. We focused on the dominant deciduous and coniferous defoliating insects[32] of boreal and hemiboreal forests[33]: European gypsy moth, forest tent caterpillar, spruce budworm (*Choristoneura fumiferana*), and

jack pine budworm (*Choristoneura pinus*). We also included two species that first appeared in our study region between 2008 and 2010: aspen two-leaf tier (*Enargia decolor*) and Bruce spanworm (*Operophtera bruceata*). The larvae of all these species typically emerge in May and feed until June or late July[32]. After feeding, larvae pupate before emerging as adults that deposit eggs—to overwinter—before dying. Together, these defoliating insects annually covered 23-times more of our study region than all bark and wood-boring beetles combined (see Supplementary Fig. 1). To our knowledge, our study of how insect outbreaks impact freshwater C and N dynamics is the most extensive spatially, temporally, and in terms of defoliator species. Previous studies have focused at most on 1–2 defoliation events, all with single insect species, making it difficult to extract wider generalities[34–38]. Long-term studies are necessary[39] to separate within- and among-year environmental effects that can confound single defoliation studies. As northern catchments are expected to experience more insect outbreaks in the near future[40,41], our work highlights the importance of terrestrial disturbances as a control of aquatic biogeochemical cycling.

## Results

**Insect outbreaks reduce canopy cover.** Using satellite imagery from 1985 to 2016, we found insect outbreaks were associated with less forest cover before autumn leaf senescence (Fig. 1). We generated monthly averages of forest cover in each catchment from the leaf area index (LAI) using 30 m resolution Landsat imagery. As expected, if defoliating insects consumed abundant foliage, LAI across our 12 study catchments was lower by a mean of 22% (95% confidence interval [CI]: 13–32%) during the growing season when the percentage of the catchment damaged increased from 0 to 100% (Fig. 1). The reduction in LAI peaked in July (mean decrease: 24%; 95% CI: 15–33%), coinciding with peak insect consumption and frass production. LAI did not differ between outbreak and non-outbreak years in May when insect feeding is minimal or in September and October when forests

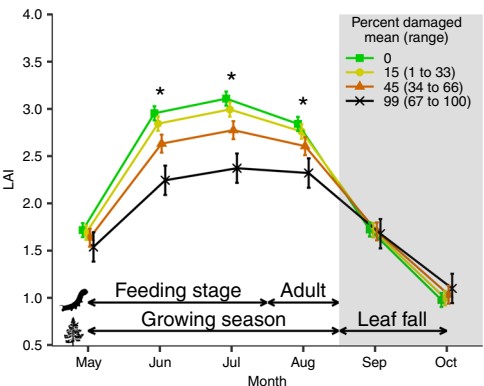

**Fig. 1 Defoliator outbreaks reduce forest cover.** We calculated the monthly leaf area index (LAI) across 12 catchments in Ontario, Canada from 1985 to 2016. For visualization purposes, monthly catchment-level averages of LAI were grouped into three equal-width bins for the proportion of the catchment area defoliated and points show the corresponding mean ± SE. However, slopes were estimated to models fitted to non-binned data with $N = 384$ per month (Supplementary Table 2). The mean value of each bin is displayed with the corresponding range in parentheses. The shaded area is the typical senescence period for the southern boreal forest. Upper arrows illustrate general life stages of phytophagous insects while lower arrows show leaf phenology. Asterisks (*) denote a statistically significant effect of the percentage of catchment damaged on LAI within a given month calculated using estimated marginal means (see Supplementary Table 2). Conditional $R_c^2 = 0.81$.

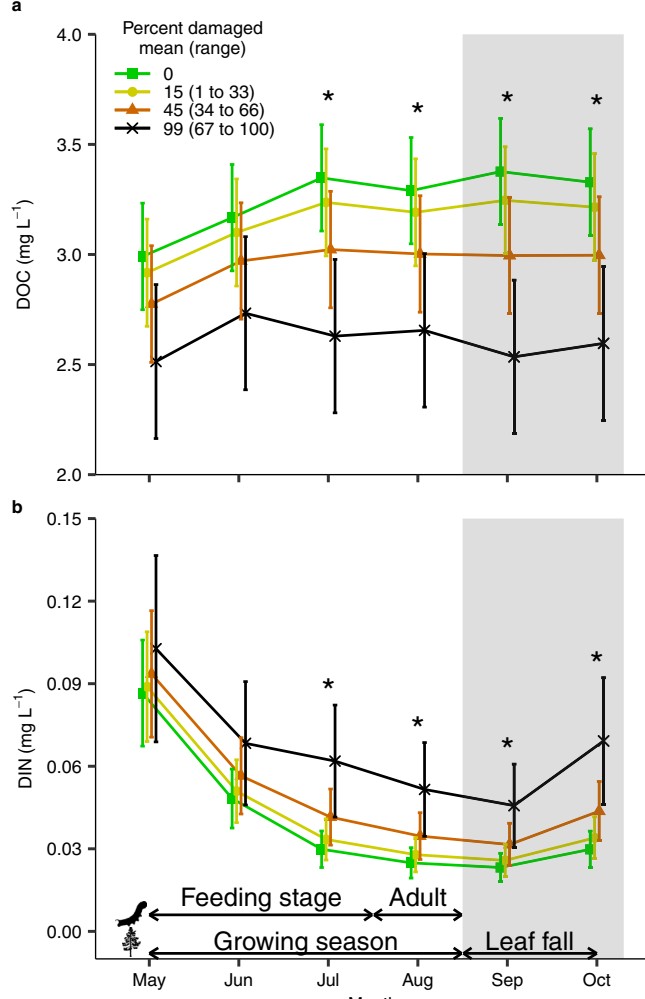

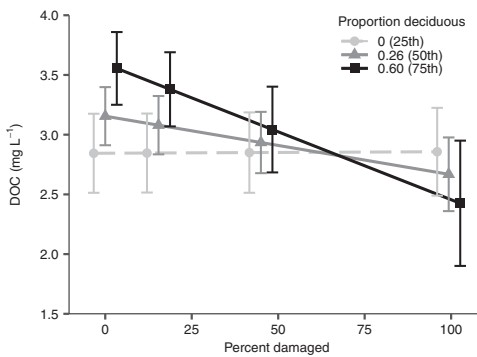

**Fig. 3 Dissolved organic carbon (DOC) declines more strongly with insect defoliation in catchments with a greater proportion of deciduous stands.** For visualization purposes, lines are mean DOC averaged across months at the 25th, 50th, and 75th percentiles for the proportion of pure deciduous forest stands in a catchment. Points (±SE) were extracted at zero and for the three equal-width bins describing the observed proportion of defoliated catchment area (0.15, 0.45, and 0.99; $n = 48$ to 1800) reported in Figs. 1 and 2. The dashed line denotes a non-statistically significant trend at low proportions of deciduous stands. Conditional $R_c^2 = 0.63$.

**Fig. 2 Less dissolved organic carbon (DOC) and more dissolved organic nitrogen (DIN) in lake water during defoliator outbreaks.** Mean (±SE) (**a**) monthly DOC and (**b**) DIN concentration averaged across 12 lakes from 1985 to 2016. For visualization purposes, monthly catchment-level averages of DOC and DIN were grouped into three equal-width bins for the percent catchment area defoliated and points show the corresponding mean ± SE. However, slopes were estimated to models fitted to non-binned data with $N = 289$–340 and 270–320 per month for DOC and DIN, respectively (Supplementary Table 2). The mean value of each bin is displayed with the corresponding range in parentheses. Shaded area is the typical period of leaf senescence for the southern boreal forests. Upper arrows illustrate the general life stages of phytophagous insects while lower arrows show leaf senescence. Asterisks (*) denote a statistically significant effect of the percentage of catchment damaged on either DOC or DIN within a given month calculated using estimated marginal means (see Supplementary Table 2). Conditional $R_c^2 = 0.63$ and 0.64 in **a** and **b**, respectively.

begin senescing (mean 95% CI for differences in LAI: −21 to 20% for May, September and October). When we compared forest cover across months with no outbreaks, LAI peaked in July and exceeded June and August values (95% CI for difference: 3–7% and 7–12%, respectively). There was no such July peak in years where outbreaks covered most of the catchment (95% CI: −1 to 13% and −9 to 5%, respectively, for a mean of 99% damage; Fig. 1).

**Biogeochemical consequences of insect outbreaks.** Lake chemistry changed during insect outbreaks consistent with defoliation

that reduces available C while increasing N inputs. When we accounted for temporal autocorrelation and variation across lakes, DOC concentrations were lower by a mean (95% CI) of 0.71 (0.19–1.23) mg L$^{-1}$ from July onwards in years with the greatest level of insect damage, as might arise if labile frass produced by defoliators primed soil microbial activity and made less C available to be leached (Fig. 2a). On a relative basis, DOC was reduced by 19–24% during this period as compared with no insect disturbance. This reduction in DOC was minimized to 0.32 (0.09–0.56) mg L$^{-1}$ when only about half (45%, Fig. 2a) of the catchment was disturbed. During years with no disturbance, lake water DOC concentrations increased from May through July and remained elevated until October (95% CI for the difference between October and both May and July: 0.15 and 0.50 and −0.18 to 0.13 mg L$^{-1}$, respectively; Fig. 2a). This trend was absent once outbreaks affected ≥47% of the catchment area, with DOC remaining consistent throughout the year (95% CI for the difference between October and both May and July: −0.004 to 0.42 and −0.23 to 0.17 mg L$^{-1}$, respectively; Fig. 2a).

DIN showed the opposite pattern to DOC. We found that, when the entire catchment was disturbed by insects, DIN concentrations increased by a mean (95% CI) of 0.03 (0.01–0.04) mg L$^{-1}$ in July when insect feeding and frass deposition typically peak (Fig. 2b). This increase in DIN during outbreak years persisted throughout the remaining ice-free season with a mean (95% CI) increase of 0.03 (0.01–0.04) mg L$^{-1}$. On a relative basis, DIN concentrations peaked in October by 134% when the percent of the area in a catchment disturbed by insects increased from 0 to 100% (Fig. 2b). N:P ratios, therefore, increased with insect outbreaks, as there were no corresponding changes in total P (Supplementary Fig. 2). Irrespective of insect defoliation, DIN concentrations always sharply decreased from May to July before increasing from September to October (Fig. 2b).

The effects of defoliator outbreaks on lake chemistry partly depended on surrounding forest composition. DOC concentrations decreased more during outbreaks in catchments with more deciduous stands (Fig. 3). At the median proportion of deciduous tree cover observed in our catchments (=0.26), an increase in the percent of the catchment damaged by insects from 0 to 100% decreased lake DOC concentrations by 0.48 (95% CI: 0.08–0.88) mg L$^{-1}$. At the upper quartile of deciduous tree cover

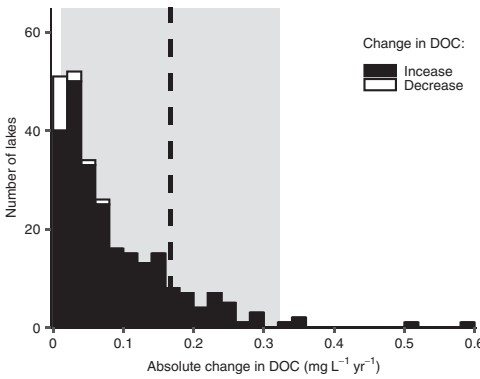

**Fig. 4 Defoliator outbreaks offset increases in lake water dissolved organic carbon (DOC) concentrations observed across northern waters.** Bars represent the number of lakes that showed either an increase (black; $n = 247$) or decrease (open; $n = 15$) in DOC between 1990 and 2016[44]. Four lakes showed no change in DOC and are omitted from the plot. The vertical dashed line is the mean decline in DOC averaged across ice-free months estimated during defoliator outbreaks over the same study period from 12 catchments in Ontario, Canada. The shaded area represents the 95% CI for the change in DOC calculated via the emmeans package.

($=0.60$), this reduction reached 1.13 (95% CI: 0.28–2.00) mg L$^{-1}$ over the range of insect damage—a 32% reduction on a relative basis (i.e., black line, Fig. 3). The proportion of coniferous stands did not change the effects of defoliators on DOC concentrations, and defoliator effects on DIN or LAI did not vary with forest composition (Supplementary Table 1).

**Effects of insect outbreaks exceed broader between-year trends.** The magnitude of changes in lake water chemistry associated with insect outbreaks can have broader biogeochemical consequences. Over the 32-year period of this study, average annual DOC concentrations have increased in the study lakes while DIN concentrations have decreased (Supplementary Fig. 3). These trends are part of broader, widely reported patterns across the boreal and north temperate regions associated with recovery from acidification and anthropogenic influences[42,43]. To contextualize the impact of defoliators relative to these inter-annual trends, we compared the change in within-year DOC concentrations from complete catchment defoliation to between-year changes in DOC. Between-year changes were calculated from 266 boreal and north temperate lakes that have been relatively undisturbed between 1990 and 2016[44]. We similarly compared within-year changes in DIN concentrations from defoliation to between-year trends, but only in our 12 study lakes because DIN was not measured in the broader spatial survey. For both these analyses, we estimated the mean annual change in DOC and DIN using our existing statistical models because only annual values were available from the larger 266-lake data set. We found defoliator outbreaks over the same years as in the wider data set (1990–2016) reduced annual ice-free mean DOC concentrations by a mean (95% CI) of 0.16 (0.01–0.32) mg L$^{-1}$, exceeding 85% of observations in the 266-lake data set on an absolute basis (Fig. 4, dashed line). This decrease exceeded the mean absolute trend in DOC across the 266-lake data set (0.08 mg L$^{-1}$ yr$^{-1}$) by two times. By contrast, we found defoliator outbreaks increased annual DIN concentrations by a mean (95% CI) of 0.012 (0.001–0.021) mg L$^{-1}$. This increase exceeded all annual trends observed in our 12-lake data set over the same study period (1985–2016) and was 12-times greater than the mean absolute annual change in those lakes (Supplementary Fig. 3).

Insect outbreaks are also a persistent and pervasive disturbance of catchments across this broader landscape, suggesting their

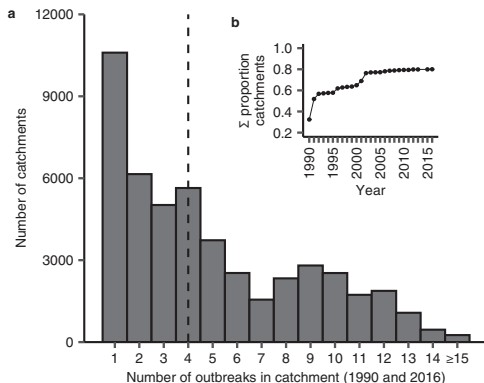

**Fig. 5 Half of all surveyed lake catchments in Ontario, Canada experience defoliator outbreaks at least every 5 years. a** Bars are the number of catchments with lakes ≥5 ha where insect defoliator outbreaks occurred between 1 and ≥15 times from 1990 and 2016 ($n = 48,286$). The dashed line is the median number of outbreaks per catchment across the 22-year interval. **b** Cumulative proportion of catchments with at least one insect outbreak between 1990 and 2016 ($n = 60,430$). Outbreaks were defined where ≥50% of the catchment area was defoliated by aerial observation.

strong effects on biogeochemical trends (e.g., Fig. 4) are ubiquitous. We counted the number of years where ≥50% of the land area was defoliated for 60,430 catchments that encompassed all lakes ≥5 ha within the 439,661 km$^2$ area of Ontario surveyed for insect outbreaks (Supplementary Fig. 4). Over the same 22-year period as the broader biogeochemical data set (1990–2016), individual lake catchments experienced a median of four defoliator outbreak events (Fig. 5a). Furthermore, within the 22 years, we found that 80% of all catchments had at least one defoliator outbreak (Fig. 5b).

## Discussion

Here we found that insect defoliator outbreaks were associated with consistent, frequent, and widespread changes in lake DOC and DIN concentrations that can mask background trends in aquatic nutrient cycling. We attributed these effects to the loss and composition of forest cover in surrounding catchments that we detected using remote sensing. The spatial and temporal breadth of our study, the most extensive to our knowledge, now advances our understanding of how defoliator outbreaks alter aquatic nutrient cycling. Most of our understanding of how defoliators impact receiving waters comes from studies of single defoliation events[34–37,45,46]. However, these studies are too short-lived to compare the magnitude of outbreak effects with broader temporal trends, such as increased DOC concentrations, as we do here. By analyzing long time series while accounting for background inter-annual trends and seasonality, our results reveal much larger effects of forest defoliators on aquatic biogeochemistry than previously appreciated.

During outbreaks, defoliators consume foliage and convert it to highly biolabile frass[26], which alters both C and N cycling. These reductions in forest leaf area were greatest during mid-summer (June to August) when insect feeding peaks[24]. C is then rapidly incorporated from frass into microbial biomass and can be respired to the atmosphere rather than accumulated in soils[47,48]. Furthermore, any highly biolabile C from frass that enters adjacent waters may be rapidly assimilated by the aquatic microbial community[49], which will further reduce lake water DOC concentrations. Less C also accumulates as foliar litter because fewer leaves/needles are shed during autumn senescence[50]. Conversely, defoliators increased lake DIN concentrations through their inefficient assimilation of foliar N and subsequent release of

soluble N-rich frass[51]. Although previous studies have shown that the addition of N-rich frass to forest soils increases soil nitrification[38,52] and eventual uptake by recovering trees[45], N leaching into aquatic ecosystems still occurs[25]. Given that northern lakes are often N-limited[53], even small additions of N will result in proportionally large increases in available N concentrations that can promote the productivity of aquatic food webs[29]. The lack of an effect of insect outbreaks on lake P concentrations is likely owing to atmospheric deposition and wetland export being the primary sources of P in our system rather than allochthonous inputs from forests[28].

Catchment tree cover, more specifically the types of foliage available to convert to frass, further dictated biogeochemical responses. Deciduous leaves are more biolabile than coniferous needles because of their higher C:N ratio, lower lignin:N ratio, and lower tannin concentration[54,55]. Increasing deciduous tree cover will therefore increase the biolability of frass and increase the incorporation of C into the soil and aquatic microbial communities. This process explains why we found lower DOC concentrations in downstream waters after defoliator outbreaks in areas with a higher proportion of purely deciduous stands. The lack of a significant interaction in our LAI analysis between tree cover type and catchment damage further supports our interpretation that the effects of defoliators arise from the processing of leaves/needles rather than vegetation consumption. In other words, defoliators collectively did not discriminate between forest types when causing damage, but the biogeochemical consequences depended on the foliar type. For DIN, defoliators assimilate so little N from leaves/needles[25], the resulting frass will likely always be high in N regardless of the differences in foliar N concentrations between deciduous and coniferous trees. However, as elevated $CO_2$ levels reduce leaf nutrient content[56], frass produced from insect outbreaks may become less nutrient-rich.

Our results contrast those from outbreaks of wood-boring insects that cause up to 85% of trees to die[57]. For example, the large amount of dead forest biomass produced by bark beetle outbreaks in the western USA, combined with the increased soil runoff resulting from less canopy cover[12,16,57], releases a pulse of OM into the surrounding environment[58]. Consequently, downstream DOC concentrations have been found to increase by 40% after bark beetle outbreaks[16]. Although defoliating and wood-boring insects can both increase N concentrations in downstream waters[12–15], these increases arise through different mechanisms between the two insect types. Relative to defoliators, wood borers release frass in lower quantities[59] and with lower C and N concentrations[60], because their food source is less nutrient-rich than foliage[61]. Wood borers are also more efficient assimilators of N because they host N-fixing gut bacteria[62], leaving their frass much less biolabile. Instead, more N is available to be leached from forests impacted by wood borers because the associated tree dieback reduces N uptake[12] and increases catchment N mineralization by adding large amounts of OM to soils[63]. When bark beetles do not cause tree mortality, the increase in downstream DIN can be weak or negligible compared with the effects of defoliators observed here[17].

Changes to terrestrial C and N inputs to lakes because of insect outbreaks have the potential to alter food web structure and composition. Lakes in our study region remain stratified until late autumn[64], during which algae continue to uptake nutrients[65,66] and respond to changes in overlying water quality associated with DOC inputs. During severe insect outbreaks, algal productivity in lakes may therefore increase in late autumn due to elevated transfer of N from forests via frass and less light attenuation by DOC from reduced litterfall inputs. Conversely, insect outbreaks could promote heterotrophic productivity by shifting nutrient limitations. The productivity of both algae and heterotrophic

bacteria in northern lakes is typically limited by P, but light-, N-, and N/P co-limitation can also occur[6,10,53]. As DIN concentrations increase during outbreak years and total P concentrations remain unchanged (Supplementary Fig. 2), lake N:P ratios will increase and basal community composition may shift towards heterotrophic bacteria that are better competitors for limited P[67]. Ongoing climate change and rising $CO_2$ concentrations will further increase terrestrial OM export to lakes by promoting soil runoff[68] and forest growth[69], respectively. These increases in terrestrial OM and the associated accumulation of DOC[70] should also promote heterotrophic growth[68] in the absence of insects. Future studies should monitor how the competitive balance between algae and bacteria shifts during insect outbreaks to determine which groups are favored by corresponding changes in nutrient cycling.

Our results have at least two implications across the wider region given ongoing global change. First, the reduction in litterfall inputs to lakes due to insects should provide a within-year pause in DOC accumulation that enables higher whole-lake productivity as more light and nutrients are available for algal uptake. The northward migration of deciduous trees associated with climate warming[71] should enhance these effects by increasing the amount of biolabile leaf litter. Second, the large range we found in the frequency of insect outbreaks may promote heterogeneity in catchment biogeochemistry across the landscape[72]. Defoliation leads to a loss of nutrients from catchments by transferring OM from forests to lakes[37], ultimately shifting the nutrient balance of C and N in lakes. More frequently defoliated catchments may therefore experience a phase shift in lake biogeochemistry relative to catchments with less-frequent outbreaks. This divergence stands to be enhanced by climate change. Forests in the study region also face an increased risk of defoliator outbreaks[73] owing to the range expansion and increased population growth of insects associated with climate change[74]. Warmer temperatures may also disrupt the relationship between trees and insects (e.g., phenology, plant defense), which will shift insects to areas more favorable to outbreaks[74]. Together, the magnitude of change in lake biogeochemistry and the variation in outbreak frequency suggest that global C and N budgets and models should now consider the effects of insect outbreaks on lake biogeochemistry across spatial and temporal scales. More generally, our results demonstrate how tracing landscape nutrient cycles requires a better understanding of the potential connections between terrestrial and aquatic ecosystems.

## Methods

**Study sites**. We focused on 12 lakes across the Algoma, Greater Sudbury, Muskoka, Temiskaming regions of ON, Canada, that have been continually monitored between 1985 and 2016. (Supplementary Table 3). All our study lakes are thermally stratify except for Wishart Lake. Lake waters were nutrient-poor, with total phosphorus concentrations between 3.1 and 6.0 µg $L^{-1}$ and N:P ratios between 32 and 104 measured via monthly surface grabs at the point of maximum depth. These low concentrations should make the lakes relatively dependent on terrestrial resource inputs[75].

Our study catchments were located across the Ontario Shield ecozone that spans boreal and hemiboreal forests[33]. Vegetation was consequently characterized by a mix of coniferous and deciduous tree stands, including eastern white pine (*Pinus strobus*), paper birch (*Betula papyrifera*), red maple (*Acer rubrum*), red pine (*Pinus resinosa*), sugar maple (*Acer saccharum*), and trembling aspen (*Populus tremuloides*). We defined the surrounding land that fed each lake (i.e., catchment) using SAGA GIS v.6.3.0[76]. A 30 m digital elevation model (DEM) obtained from the Ontario Ministry of Natural Resources and Forestry[77] was pre-processed by filling in sinks[78] before generating a flow accumulation model assuming unidirectional water flow. When the catchments of our 12 study lakes contained those of smaller upstream lakes, we removed the latter to ensure each catchment was draining only one lake.

**Insect monitoring**. We used long-term data on the extent of insect damage across Ontario collected by Natural Resources Canada—Canadian Forest Service (CFS) and the Ontario Ministry of Natural Resources and Forestry (OMNRF). Annually

since the 1940s, the CFS and OMNRF have conducted aerial flyovers of most of Ontario (total area 439,661 km$^2$), spanning between 44.3 and 52.4° latitude (Supplementary Fig. 4). This area represents two-thirds of the Ontario Shield ecozone (total area 653,368 km$^2$). Flyovers occurred immediately after the peak of defoliation events, typically in late July[79]. Trained observers in the aircraft mapped the extent and severity of any insect activity. Flight lines have varied over time but were typically 6–10 km apart, at 170 km hr$^{-1}$ and at a height of 360–600 m above ground level[79]. Insect outbreaks were considered aerially mappable when moderate to severe defoliation was observed, defined by the absence of at least 25% of foliage cover[79]. As 90% of disturbance events were classified as moderate–severe in our data set, there was effectively no variation in damage severity. We, therefore, calculated the percent of each study catchment that was damaged by insects to quantify insect disturbance in subsequent analyses. Our analysis focused on surveys from 1985 onward to coincide with the available lake chemistry data. Using the sf package v.0.9.8[80] in R v.4.1.0[81], we clipped polygons of annual insect damage to the catchment boundaries of each lake.

To determine the spatial and temporal extent of insect outbreaks across Ontario, more generally, we quantified the number of catchments in each year where outbreaks of defoliating insect species covered ≥50% of the catchment area. We selected 50% as a threshold to limit our analysis to only larger-scale outbreaks thereby ensuring small events within the catchment were not overinflating our estimates of spatial extent. We used catchments delineated by the OMNRF for all lakes ≥5 ha[82] that were within the area surveyed for insects (Supplementary Fig. 4). The proportion of each catchment that was damaged was calculated as described above but we included all defoliating insect species surveyed, not just the species present in our original 12 study catchments (Supplementary Table 4). When outbreak polygons overlapped (i.e., an outbreak of multiple species within a single year), we combined overlapping sections to ensure the damaged area within the catchment was not overinflated by counting polygons twice.

**Forest cover during insect outbreaks**. We quantified if outbreaks reduced vegetation cover in the study catchments using LAI. LAI measures forest canopy cover of deciduous and coniferous trees and is a better predictor of forest cover compared with the widely used normalized difference vegetation index (NDVI) because it is less susceptible to over-saturation at higher vegetation density and greenness[83]. However, LAI imagery has a coarser resolution (500 m$^2$) and is only available after 1999. Therefore, we calculated LAI from Landsat NDVI that is available over our entire study period and has a finer spatial resolution. We used the 30-m, 16-day interval Landsat composite available from Robinson et al.[84] from 1985 to 2016. As insect outbreaks are often localized disturbances over weeks[32], noise in NDVI during the feeding window can mask underlying signals of defoliation. Smoothing functions can improve signal-to-noise ratio of NDVI time series by removing random errors associated with erroneous geo-location, angular variations, clouds, and atmospheric disturbances[85], and have been used to measure the reductions in NDVI from insect outbreaks[86,87]. Prior to analysis, we therefore smoothed the NDVI time series with the Savitsky-Golay algorithm of TIMESAT v.3.3[88]. We then averaged NDVI across all pixels in each catchment for each month.

To convert NDVI values to LAI, we fitted a non-linear model between the Landsat NDVI composite and MODIS LAI (MODIS/006/MCD15A3H) across our catchments in 2016. We selected 2016 as this was the most recent year with no insect outbreaks within the prior two years (Supplementary Fig. 5). MODIS LAI imagery was filtered to exclude cloudy pixels and Landsat imagery was downscaled from 30 m$^2$ to 500 m$^2$ by taking the median value[89] after removing negative and no data values. The Landsat imagery was then re-sampled to match the MODIS LAI pixels before computing average annual values[90]. LAI and NDVI are non-linearly related[91], so we fitted a non-linear model using the nlsLM function[92] in R. The resulting model was then used to convert the average monthly NDVI values described above to LAI (Supplementary Fig. 6).

We also calculated the proportion of each catchment that was covered by coniferous or deciduous tree cover using the Ontario Land Cover Compilation v.2.0[93]. Land cover was clipped to each catchment and the proportion of total forested area that was classified as solely deciduous or coniferous was extracted. Using only forest area ensured we were not including land cover types that insects would not feed on, e.g., barren, water, grasses. We also excluded mixed and sparse forests from our analysis because there was no indication of the ratio between coniferous and deciduous trees within these classifications.

**Lake chemistry**. We obtained DOC, DIN (nitrates plus ammonium), and TP concentrations for ice-free months (May to October) for each lake to determine how they responded to insect outbreaks. Water was collected with surface grabs near the deepest point of each lake. Grab samples were the most consistent record of water chemistry across all our study regions, and so used for consistency. When multiple samples were collected per month, we averaged values. DOC and DIN were measured with automated colorimetry using standard protocols by the Ontario Ministry of the Environment[94] for lakes in the Greater Sudbury, Muskoka, and Temiskaming regions, and by the CFS at the Great Lakes Forestry Centre for lakes in the Algoma region. In brief, DOC was measured by acidifying samples and flushing them with nitrogen gas to remove inorganic carbon. Organic carbon was then oxidized to $CO_2$ and measured colorimetrically with a phenolphthalein indicator[95]. These methods have remained consistent over the study period to

ensure the resulting DOC and DIN concentrations remain comparable through time.

To contextualize the within-year changes in lake chemistry from insect outbreaks, we compared the magnitude of our results with between-year trends in DOC and DIN. For DOC, we collated data from 266 lakes in the International Cooperative Programme on Assessment and Monitoring of the Effects of Air Pollution on Rivers and Lakes (ICP Waters) Programme that monitors water chemistry across the boreal and north temperate. Catchments monitored by ICP Waters are selected to avoid disturbance[96], and their broad spatial coverage makes it ideally suited to track trends in lake chemistry more generally. For each lake, we calculated the absolute Theil-Sen slope for the inter-annual trend in DOC concentrations between 1990 and 2016. This approach provides a non-parametric estimate of the slope of a linear regression line fitted through two variables that is more robust to outliers. We compared these values to the change (i.e., slope) in mean annual DOC concentrations associated with complete catchment disturbance. The effect of insect damage was estimated from mixed-effects models fitted to our 12 study lakes between 1990 and 2016 (see below for fitting procedure). For DIN, we calculated between-year trends from our 12 study lakes because the wider ICP waters data set does not sample DIN. Like for DOC, we calculated the absolute Theil-Sen slope for the temporal trend in mean annual DIN in each lake and compared it to the estimated within-year effect of insect defoliation.

**Statistical analyses**. To test the impacts of insect outbreaks on LAI, DOC, DIN, TP, and DIN:TP in the 12 study lakes, we fitted separate mixed-effects models to each variable using the nlme package v.3.1.152 in R[97]. The percent of catchment disturbed by defoliators was included as a fixed effect. As defoliation is generally limited to May through July, and terrestrial OM inputs to lakes typically peak in autumn (Sept/Oct), we also let the effect of disturbance vary with sampling month (i.e., statistical interaction between months and disturbance). Doing so allowed us to determine if the effects of insect defoliators were specific to certain months during the ice-free season. We also tested if the effects of insect defoliation depended on foliar type by including an interaction between percent damaged and forest cover (deciduous and coniferous). Finally, we included forest cover as the main effect in our model. We accounted for variation among sites, such as those associated with differences in disturbance history, climate, timing of stratification events, and soil type, by including catchment identity as a random effect. Including catchments as a random effect ensures that our model output represents the overall effect of insect outbreaks on lake biogeochemistry while incorporating the variation across catchments. We also accounted for temporal autocorrelation in each response by estimating a continuous first-order autoregressive (CAR1) error structure within each lake. The CAR1 structure ensures that the observed trends in our response variables were not reflecting other seasonal or long-term processes, e.g., brownification or forest greening. The emmeans package v.1.6.0 in R[98] was used to calculate 95% CIs and compare between groups.

## Data availability

MODIS/006/MCD15A3H is available from Google Earth Engine. Landsat composite data are available through https://ndvi.ntsg.umt.edu. The International Cooperative Programme for assessment and monitoring of the effects of air pollution on rivers and lakes data set (ICP Waters) is available at http://www.icp-waters.no. The lake chemistry, insect outbreak, and catchment data used in this study have been deposited in the Zenodo database (https://doi.org/10.5281/zenodo.5517857).

## Code availability

The code is provided in the supplementary file entitled Supplementary Code.

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

## Acknowledgements

We thank Jocelyne Heneberry, Ron Ingram, Andrew Paterson, Michael McTavish, Brian Kielstra, Laura Bentley, and Erika Freeman for their guidance in collating and analyzing our data, and W. Wyatt Hoback and two anonymous reviewers for comments that improved an earlier draft. We would also like to thank CFS, OMNRF, the Inland Waters Unit of the Ontario Ministry of the Environment, Conservation and Parks, and the Climate Change and Atmosphere Pollutants program from Environment and Climate Change Canada for providing us with access to the insect disturbance and lake chemistry data used in this publication under the Open Government License—Canada. This work was supported by Natural Environment Research Council grant (NE/L006561/1) to A.J.T. and the Ontario Centres of Excellence (OCE/27649) and the Natural Sciences and Engineering Research Council of Canada (NSERC/509182-17) to J.M.G.

## Author contributions

S.G.W., J.M.G., and A.J.T. conceived the study. S.G.W. collated the data sets with assistance from R.E.F. for the insect outbreak data. S.K. helped analyze LAI data. E.J.S.E., J.M.G., and J.A.R. assisted with the acquisition of lake chemistry data. S.G.W. analyzed the data and wrote the manuscript with input from all authors.

## Competing interests

The authors declare no competing interests.
