## [Peer Review File · Nature Communications]

REVIEWER COMMENTS

Reviewer #1 (Remarks to the Author):

Review of "Forest defoliator outbreaks disrupt nutrient cycling in northern waters" by Woodman and others

In this manuscript, the authors attempt to describe the universal characteristics of the effects of defoliator outbreaks on forest biogeochemical cycles. In particular, they focus on the effects of defoliator outbreaks on the discharge of DOC and DIN from forests to lakes, using long-term observational data from 12 lake basins in Ontario, Canada. It is commendable that the authors were able to robustly extract the effects of defoliation with 32 years of long-term data, and this finding is worthy of publication.

However, I do not believe that the paper is ready for publication at this time because it contains several problems, as described below.

General comments:

As indicated in the title, the authors argue that the findings from the forest-lake dataset of one particular region, Ontario, represent a universal characterization of the boreal forest. In reality, however, the vast ecosystems of the boreal forest are subject to a wide variety of hydrological, climatic, and biogeochemical conditions. Therefore, they would need to consider the possible spatial and temporal frequency of their effects in the entire boreal forest before saying the universality of the phenomena they are discussing.

They compare data from other lakes in the "northern hemisphere" with data in Ontario. However, the details of the data used for comparison are not properly described, so it is difficult to determine whether they are adequate to emphasize the impact of the defoliator outbreak, as the author concludes. Did the 266-lake dataset not include data that was affected by defoliation?

2. Do forest defoliator outbreaks really disrupt nutrient cycling in northern waters? Even in natural forests where there are no defoliator outbreaks, defoliation occurs every year, and nutrient cycling continues as the litter decomposes. Defoliator outbreaks appear to add the function to accelerate this defoliation and litter decomposition. The nutrient cycle may be disturbed, but it may not be destroying the forest nutrient cycle. For example, if forest dynamics encompassing the effects of a deciduous person outbreak every few years were to continue at a steady-state, I don't think this would be disruption. It could simply be defined as the occurrence of phase shifting.

In the conclusions section, the authors list several possible outcomes of the increased impact of defoliator outbreaks, citing studies from several other regions. For example, increased DIN export from catchments and depletion of nitrogen in forests. However, these simply assume that the qualitative trends indicated by the current situation will be accelerated, and do not provide sufficient evidence to support that this will simply proceed.

Comments on individual points:

In Abstract:

"Compared to regional trends, a single insect outbreak could entirely pause DOC increases observed across 266 northern hemisphere lakes from 1990-2016. Similarly, insect outbreaks exceeded local trends in annual DIN by 30-times, on average."

It is not so easy to understand these sentences. In reading the text, I do not see any discussion that concludes this. The second sentence is incomprehensible. [insect outbreak] and [local trends in annual DIN] are compared. Please add the necessary words to complete this sentence.

In Figures 1, 2 and Sup. Figure 2, the intensity of defoliation (% damage) is presented as mean and variance for each intensity class, while changes in DOC and DIN concentrations and the year in which defoliation occurred are presented for each watershed in Sup. Figures 3 and 5. In the end, it is not directly

clear in which basin the defoliator outbreak occurred and at what intensity, giving the impression that the correlations are not clear. In particular, with regard to DIN export, there are cases where DIN concentrations were high even in watersheds that were not so intensive outbreak (Little Turkey and Wishart) in Sup. Figure 5. The causal relationship discussed in the text cannot be certainly verified catchment by catchment.

Reviewer #2 (Remarks to the Author):

The manuscript combines aerial insect herbivory mapping data with NDVI/LAI imagery and long-term lake water chemistry data to test the importance of insect defoliator outbreaks for biogeochemical cycles in forest catchments. The authors report significantly lower DOC and higher DIN in lake waters during years of defoliator outbreaks and demonstrated that DOC declined more strongly in catchments dominated by deciduous trees. The main data covers 12 boreal lake catchments over 32 years. Additionally, larger spatial scale data is utilized to compare the defoliator outbreak effects on temporal trends in lake water chemistry, for instance. The experimental set up and statistical analyses appear appropriate for their conclusions. The manuscript is very interesting and clearly written, and I only have some minor comments, as outlined below.

Line 91: Just noted that Jepsen et al 2008 could be a good reference here.

Jepsen et al. 2008. Climate change and outbreaks of the geometrids *Operophtera brumata* and *Epirrita autumnata* in subarctic birch forest: evidence of a recent outbreak range expansion. *Journal of Animal Ecology* 77:257-264

Figures 1-2 (and Supplementary fig 2): I would present n for each dot in these figures and refer to the Supplementary Table 1. If n varies, maybe a range is sufficient. In addition, it is said that * denotes a statistical difference between "different damage groups" but the exact statistics are not presented anywhere. I'm not sure whether I interpret the statistics of Supplementary table 1 correctly but with a quick look it seems that for DOC (Fig 2a) the statistical differences presented in Figure with * are contrasting with the results provided in Supplementary Table 1.

Figure 3: Would this figure be too fuzzy if you present all individual observations in it? In a background using very pale colours? Please at least present n for each dot here as well.

Supplementary Figure 3: Would this figure be more informative if you use different colour or symbol for the years of defoliator outbreaks?

Figure 4: This is great visualization of the magnitude of herbivory effect. n increased and decreased adds up only to 262 – so was there no change at all for 4 lakes? It is not very clear to me how did you determine the absolute change in DOC, maybe it could be clarified in methods.

Figure 5: Can you see any trends for increased frequency of outbreaks from this data (or from complete annual herbivory mapping data collected since 40s)?

Line 312-315, 323-325: Are there any studies on why some catchments are, or would be in the future, more frequently disturbed?

Methods/insect damage: It is told here that outbreaks were aerially mappable when moderate to severe defoliation was observed, defined by the absence of at least 25% of foliar cover, and that extent and severity of insect activity was mapped. Thus, it seems that some outbreak areas have been classified as more severely defoliated than others but did you take the severity of defoliation into account when the proportion of disturbance was calculated?

Supplementary Table 1: Treed should probably be trees, right?

Supplementary Figure 1, line 731: should it be from 2004 to 2004?

Reviewer #3 (Remarks to the Author):

The manuscript "Forest defoliator outbreaks disrupt nutrient cycling in northern waters" by Woodman et al. presents analysis of long-term trends in outbreaks of defoliating insects and inputs to northern watersheds. The authors present analysis of data from more than 10,000 watersheds over a period of nearly 30 years. The authors found that the type of tree (gymnosperms vs. angiosperms) affected the amount of dissolved carbon released by defoliators.

Overall, the manuscript is well-written and with climate change and increasing frequency of gross-tissue removing (defoliating) insects, the results of this study will be valuable to a wide readership and will provide an excellent baseline for follow-up studies.

Major concerns:

Although I understand the authors use of colors to represent forest colors, the figure colors need to be updated and different symbols used (Figure 2) because the barely different colors do not stand out. Some information should be added about the types of lakes. Are these lakes deep and thermally stratified? Is there constant uptake by algae. Introduction should include more about phosphorus nitrogen ratios and algal response.

Lines 300-303. Epilemnetic DOC suggests lakes (at least some) are thermally stratified. The authors need to address language in the discussion as their thoughts are unclear. Are the results observed in water quality similar across lakes or a source of variation. The timing of defoliation should be a period during stratification and leaf fall should occur as the stratification breaks down.

It is surprising that the authors did not observe changes in other nutrients associated with outbreaks of defoliators. Phosphorus is a key nutrient which appeared unaffected. Why would this be the case?

Although the authors cite a large number of papers, I feel that a number of important papers are not cited. Perhaps they could pair down their current references and add some of the other papers here.

With climate change, plants are producing more foliage but in natural conditions, the foliage is often nutrient poor. The authors should reference changes in plant nutrition associated with elevation in CO₂. There is a growing body of literature but the paper by Newall et al. 2018 Aquatic Sciences 80: 27 and references within would help with interpretations. In addition, Lapierre et al. 2013 Nature Communications is very relevant to interpreting changes in aquatic DOC that may also be influenced by atmospheric CO₂.

Recently, DOC (a main component measured by the authors) has been linked to browning regardless of other factors. See Kritzberg et al. 2020. Ambio 49: 375-390. The authors should consider if increasing DOC in northern climates is affecting water clarity regardless of bacteria.

The authors need to better connect the results of the study to the broad implications of accumulating DOC (lines 308-309) and being paused by defoliator outbreaks. In the context of overall lake health, are defoliators good?

Minor comments:

Line 34: macronutrients, including carbon

Line 39: by affecting (not controlling)

Line 75: sites because

Line 168: - a 45% (typo?)

Line 238: nutrient cycling in temperate zones

Line 248: nutrient rich frass has some important nutrients, nitrogen not phosphorus for example

Line 258: even small additions of N will result in proportionally large increases in concentrations that will be biologically important. This is unclear. Concentrations of available Nitrogen? Biomagnification? This seems important but very unclear.

Line 282: mechanisms by which they achieve that differ. Unclear what is meant.

Line 301: The connection of DOC to decreased bacteria favoring pelagic algal species needs to be better explained or cut from discussion.

W. Wyatt Hoback

Reviewer #1 (Remarks to the Author):

Review of “Forest defoliator outbreaks disrupt nutrient cycling in northern waters” by Woodman and others

In this manuscript, the authors attempt to describe the universal characteristics of the effects of defoliator outbreaks on forest biogeochemical cycles. In particular, they focus on the effects of defoliator outbreaks on the discharge of DOC and DIN from forests to lakes, using long-term observational data from 12 lake basins in Ontario, Canada. It is commendable that the authors were able to robustly extract the effects of defoliation with 32 years of long-term data, and this finding is worthy of publication.

We are glad the Reviewer finds our results worthy of publication.

However, I do not believe that the paper is ready for publication at this time because it contains several problems, as described below.

We thank the Reviewer for these insightful comments, which we address in detail below.

General comments:

As indicated in the title, the authors argue that the findings from the forest-lake dataset of one particular region, Ontario, represent a universal characterization of the boreal forest. In reality, however, the vast ecosystems of the boreal forest are subject to a wide variety of hydrological, climatic, and biogeochemical conditions. Therefore, they would need to consider the possible spatial and temporal frequency of their effects in the entire boreal forest before saying the universality of the phenomena they are discussing.

We did not intend to suggest that our findings can be universally applied across the boreal forest. Although the title did not argue for this extrapolation, the Abstract did, and we think this led to the Reviewer's comments.

Changed on lines 28-30

“As insect defoliator outbreaks occur at least every 5 years across a wider 439,661 km² boreal ecozone **of Ontario**, we suggest they are an underappreciated driver of biogeochemical cycles in forest catchments **of this region**.”

We could find no other instances in the Discussion where we suggested that our findings were a universal characterisation of the boreal forest so have made no further changes.

They compare data from other lakes in the "northern hemisphere" with data in Ontario. However, the details of the data used for comparison are not properly described, so it is difficult to determine whether they are adequate to emphasize the impact of the defoliator outbreak, as the author concludes. Did the 266-lake dataset not include data that was affected by defoliation?

Changed on lines 194-203:

“To contextualize the impact of defoliators relative to these inter-annual trends, we compared the change in within-year DOC concentrations from complete catchment defoliation to between-year changes in DOC. Between-year changes were calculated from 266 lakes across

the northern hemisphere that have been relatively undisturbed between 1990 and 2016⁴⁴. We similarly compared within-year changes in DIN concentrations from defoliation to between-year trends, but only in our 12 study lakes because DIN was not measured in the broader spatial survey. For both these analyses, we estimated the mean annual change in DOC and DIN using our existing statistical models because only annual values were available from the larger 266-lake dataset.”

Added to lines 449-466

“To contextualize the within-year changes in lake chemistry from insect outbreaks, we compared the magnitude of our results to between-year trends in DOC and DIN. For DOC, we collated data from 266 lakes in the International Cooperative Programme on Assessment and Monitoring of the Effects of Air Pollution on Rivers and Lakes (ICP Waters) Programme that monitors water chemistry across the northern hemisphere. Catchments monitored by ICP Waters are selected to avoid disturbance⁹⁶, and their broad spatial coverage makes it ideally suited to track trends in lake chemistry more generally.”

The methodology from ICP Waters explicitly states that lakes are selected to avoid disturbances, and this should include complete defoliation. Furthermore, the comparison of our results to the ICP Waters data is not meant to act as a control, i.e. comparing lakes with disturbance to lakes that experience no disturbance. Rather, the comparison is intended to contextualize our results and show that the magnitude of the within-year change in lake chemistry following an outbreak is larger than the between-year trends in lake chemistry. We hope the changes described above now make these points much clearer.

2. Do forest defoliator outbreaks really disrupt nutrient cycling in northern waters? Even in natural forests where there are no defoliator outbreaks, defoliation occurs every year, and nutrient cycling continues as the litter decomposes. Defoliator outbreaks appear to add the function to accelerate this defoliation and litter decomposition. The nutrient cycle may be disturbed, but it may not be destroying the forest nutrient cycle. For example, if forest dynamics encompassing the effects of a deciduous insect outbreak every few years were to continue at a steady-state, I don't think this would be disruption. It could simply be defined as the occurrence of phase shifting.

In retrospect, we agree that “disrupt” is a poor choice of words. As the Reviewer rightly points out, insect outbreaks are not “destroying” background trends in nutrient cycling – they are disturbing the system away from the within-year norm. We have therefore changed the two instances where we used “disrupt” to “alter” as suggested by the Reviewer.

Changed line 1

“Forest defoliator outbreaks **alter** nutrient cycling in northern waters”

Changed line 19

“Insect defoliators **alter** biogeochemical cycles from land into receiving waters by consuming terrestrial biomass and releasing bioavailable frass.”

In the conclusions section, the authors list several possible outcomes of the increased impact of defoliator outbreaks, citing studies from several other regions. For example, increased DIN export from catchments and depletion of nitrogen in forests. However, these simply assume that the qualitative trends indicated by the current situation will be accelerated, and do not provide sufficient evidence to support that this will simply proceed.

We have now removed the text on the original lines 315-325 that listed possible outcomes of future defoliator impacts as we agree the evidence was limited to support these points. We now provided a more focused conclusion that is better supported by the literature.

Added lines 325-343

“Our results have at least two implications across the wider region given ongoing global change. First, the reduction in litterfall inputs to lakes from insects should provide a within-year pause in DOC accumulation that enables higher whole-lake productivity as more light and nutrients are available for algal uptake. The northward migration of deciduous trees associated with climate warming⁷¹ should enhance these effects by increasing the amount of biolabile leaf litter. Second, the large range we found in the frequency of insect outbreaks may promote heterogeneity in catchment biogeochemistry across the landscape⁷². Defoliation leads to a loss of nutrients from catchments by transferring OM from forests to lakes³⁷, ultimately shifting the nutrient balance of C and N in lakes. More frequently defoliated catchments may therefore experience a phase shift in lake biogeochemistry relative to catchments with less frequent outbreaks. This divergence stands to be enhanced by climate change. Forests in the study region also face an increased risk of defoliator outbreaks⁷³ due to the range expansion and increased population growth of insects associated with climate change⁷⁴. Warmer temperatures may also disrupt the relationship between trees and insects (e.g., phenology, plant defence) which will shift insects to areas more favourable to outbreaks⁷⁴. Together, the magnitude of change in lake biogeochemistry and the variation in outbreak frequency suggest that global C and N budgets and models should now consider the effects of insect outbreaks on lake biogeochemistry across spatial and temporal scales.”

Comments on individual points:

In Abstract:

“Compared to regional trends, a single insect outbreak could entirely pause DOC increases observed across 266 northern hemisphere lakes from 1990-2016. Similarly, insect outbreaks exceeded local trends in annual DIN by 30-times, on average.”

It is not so easy to understand these sentences. In reading the text, I do not see any discussion that concludes this. The second sentence is incomprehensible. [insect outbreak] and [local trends in annual DIN] are compared. Please add the necessary words to complete this sentence.

Changed on lines 25-28

“**Within-year changes in DOC from insect outbreaks exceeded between-year trends across a larger dataset of 266 northern hemisphere lakes from 1990 to 2016.** Similarly,

within-year increases in DIN from insect outbreaks exceeded local, between-year changes in DIN by 12-times, on average.”

In Figures 1, 2 and Sup. Figure 2, the intensity of defoliation (% damage) is presented as mean and variance for each intensity class, while changes in DOC and DIN concentrations and the year in which defoliation occurred are presented for each watershed in Sup. Figures 3 and 5. In the end, it is not directly clear in which basin the defoliator outbreak occurred and at what intensity, giving the impression that the correlations are not clear. In particular, with regard to DIN export, there are cases where DIN concentrations were high even in watersheds that were not so intensive outbreak (Little Turkey and Wishart) in Sup. Figure 5. The causal relationship discussed in the text cannot be certainly verified catchment by catchment.

The Reviewer is correct that it was previously difficult to verify the causal relationship among outbreak events, outbreak intensity, and lake chemistry for each catchment. We have now updated Supplementary Figures 3 and 5 to include information on outbreak events and outbreak intensity, respectively to aid in the interpretability of disturbance patterns and biogeochemical changes catchment by catchment.

Changes made on lines 783-789 to Supplementary Figure 3 to show what year outbreaks occurred within each catchment (when chemistry data were not missing).

“Supplementary Figure 3. Temporal trends in water chemistry in the 12 study lakes from 1985 to 2016. Average ice-free (May to October) (a) DOC concentrations increased while (b) DIN concentrations decreased. Theil-Sen’s slopes and p values represent the rate of change and significance in (a) DOC (mg L^{-1}) or (b) DIN ($\mu\text{g L}^{-1}$) for each catchment. Blue points indicate years with outbreaks while red represents no outbreaks.”

Changes made to Supplementary Figure 5 to show the intensity of disturbance within each catchment. Lines 787-801

“Supplementary Figure 5. History of insect outbreaks within each catchment. Green tiles represent the proportion of catchment disturbed while white tiles show years where no outbreaks occurred.”

Reviewer #2 (Remarks to the Author):

The manuscript combines aerial insect herbivory mapping data with NDVI/LAI imaginary and long-term lake water chemistry data to test the importance of insect defoliator outbreaks for biogeochemical cycles in forest catchments. The authors report significantly lower DOC and higher DIN in lake waters during years of defoliator outbreaks and demonstrated that DOC declined more strongly in catchments dominated by deciduous trees. The main data covers 12 boreal lake catchments over 32 years. Additionally, larger spatial scale data is utilized to compare the defoliator outbreak effects on temporal trends in lake water chemistry, for instance. The experimental set up and statistical analyses appear appropriate for their conclusions. The manuscript is very interesting and clearly written, and I only have some minor comments, as outlined below.

We thank the Reviewer for their very positive feedback and have tried to address each of their minor comments below.

Line 91: Just noted that Jepsen et al 2008 could be a good references here.

Jepsen et al. 2008. Climate change and outbreaks of the geometrids *Operophtera brumata* and *Epirrita autumnata* in subarctic birch forest: evidence of a recent outbreak range expansion. *Journal of Animal Ecology* 77:257-264

Thank you review for recommending this excellent addition. We have cited it as suggested.

Change made to lines 92-94

“As northern catchments are expected to experience more insect outbreaks in the near future^{40,41}...”

Figures 1-2 (and Supplementary fig 2): I would present n for each dot in these figures and refer to the Supplementary Table 1. If n varies, maybe a range is sufficient.

We have added n for each dot in Figures 1, 2, 3, and Supplementary Figure 2. These figures summarise statistical models fit to continuous insect damage within each month. To improve visualisation, we used a semi-arbitrary binning of the continuous damage values (as noted in the figure captions). However, plotting n for the bins would over emphasise the visualization and not the number of data points actually used to determine the effects in the statistical models we reported. Therefore, we have elected to report the n range for each month. This approach allows the reader to understand the number of data points used in the statistical models to estimate changes in our response variables over the range of insect damage.

In addition, it is said that * denotes a statistical difference between “different damage groups” but the exact statistics are not presented anywhere. I’m not sure whether I interpret the statistics of Supplementary table 1 correctly but with a quick look it seems that for DOC (Fig 2a) the statistical differences presented in Figure with * are contrasting with the results provided in Supplementary Table 1.

The Reviewer is correct that the statistical differences in Supplementary Table 1 do not match the asterisk in Figures 1, 2 and 3. Supplementary Table 1 summarises the overall model and not the within-month comparisons presented in the figures. In other words, the significance in Supplementary Table 1 indicates a difference from the intercept of the model

and not within group (i.e. month) effects. We have added a new table, Supplementary Table 2, to report the within-month effects for disturbance on LAI, DOC, and DIN. These values were computed using the emmeans package in R. We have also updated the caption in Supplementary Table 1 to better articulate that we present the model summary.

Added lines 117-121

For visualization purposes, monthly catchment-level averages of LAI were grouped into three equal-width bins for the proportion of catchment area defoliated and points show corresponding mean \pm SE. However, slopes were estimated to models fitted to non-binned data with $N = 384$ per month (Supplementary Table 2).

Added lines 124-126

“Asterisks (*) denote a statistically significant effect of the percentage of catchment damaged on LAI within a given month calculated using estimated marginal means (see Supplementary Table 2).”

Added lines 157-161

For visualization purposes, monthly catchment-level averages of DOC and DIN were grouped into three equal-width bins for the percent catchment area defoliated and points show corresponding mean \pm SE. However, slopes were estimated to models fitted to non-binned data with $N = 289$ to 340 and 270 to 320 per month for DOC and DIN, respectively (Supplementary Table 2).

Added lines 164-166

“Asterisks (*) denote a statistically significant effect of the percentage of catchment damaged on either DOC or DIN within a given month calculated using estimated marginal means (see Supplementary Table 2).”

Added lines 771-775

For visualization purposes, monthly catchment-level averages of TP and DIN:TP were grouped into three equal-width bins for the percent catchment area defoliated and points show corresponding mean \pm SE. However, slopes were estimated to models fitted to non-binned data with $N = 296$ to 358 and 240 to 312 per month for TP and DIN:TP, respectively (Supplementary Table 2).

Added lines 779 -782

“Asterisks (*) denotes a statistically significant effect of the percentage of catchment damage on monthly DIN:TP within a given month calculated using estimated marginal means (see Supplementary Table 2).”

Change to lines 810-813

“Supplementary Table 1. Overall model summary outputs for the effects of insect damage, month, and forest cover on LAI, DOC and DIN across our 12 study catchments/lakes. All continuous independent variables were scaled to a mean of zero and standard deviation of

one to compare effect sizes. Model estimates and 95% confidence intervals (in parentheses) are displayed along with the associated p-values. Significant p-values indicate a statistical difference from the model intercept. “

Table added to lines 815-818

Supplementary Table 2. Within-month effects of percentage of catchment damaged on LAI, DOC, DIN, TP, and DIN:TP. Values were computed using the *emmeans* package in R. Modelled estimates and 95% confidence intervals (in parentheses) are displayed along with associated p-values. Bolded p-values indicate statistically significant effects, i.e. different from zero.

Month	LAI		DOC		DIN		TP		DIN:TP	
	Estimates	p	Estimates	p	Estimates	p	Estimates	p	Estimates	p
May	-0.18 (-0.46 - 0.10)	0.208	-0.48 (-1.00 - 0.04)	0.071	0.17 (-0.33 - 0.67)	0.501	-0.001 (-0.22 - 0.21)	0.992	0.14 (-0.42 - 0.69)	0.623
June	-0.71 (-0.99 - -0.43)	<0.001	-0.44 (-0.95 - 0.08)	0.096	0.35 (-0.15 - 0.85)	0.166	-0.01 (-0.22 - 0.20)	0.909	0.34 (-0.21 - 0.89)	0.222
July	-0.74 (-1.02 - -0.46)	<0.001	-0.72 (-1.24 - -0.21)	0.006	0.73 (0.24 - 1.23)	0.004	0.04 (-0.17 - 0.25)	0.687	0.72 (0.17 - 1.27)	0.010
Aug	-0.52 (-0.80 - -0.24)	<0.001	-0.64 (-1.16 - -0.12)	0.015	0.73 (0.23 - 1.24)	0.004	-0.03 (-0.25 - 0.40)	0.745	0.71 (0.16 - 1.27)	0.012
Sep	-0.05 (-0.31 - 0.24)	0.753	-0.85 (-1.37 - -0.33)	0.001	0.68 (0.18 - 1.19)	0.008	0.0002 (-0.21 - 0.21)	0.998	0.65 (0.09 - 1.20)	0.023
Oct	0.12 (-0.16 - 0.40)	0.390	-0.74 (-1.26 - -0.22)	0.005	0.85 (0.34 - 1.36)	0.001	0.05 (-0.17 - 0.26)	0.673	0.77 (0.21 - 1.33)	0.008

Figure 3: Would this figure be too fuzzy if you present all individual observations in it? In a background using very pale colours? Please at least present n for each dot here as well.

Yes, including the individual observations results in a fuzzy figure that is difficult to interpret. We have included the figure here to illustrate our point.

Supplementary Figure 3: Would this figure be more informative if you use different colour or symbol for the years of defoliator outbreaks?

We have updated Supplementary Figure 3 by colouring points to indicate disturbance history (Lines 783 – 789).

Supplementary Figure 3. Temporal trends in water chemistry in the 12 study lakes from 1985 to 2016. Average ice-free (May to October) (a) DOC concentrations increased while (b) DIN concentrations decreased. Sen's slopes and p values represent the rate of change and significance in (a) DOC (mg L^{-1}) or (b) DIN ($\mu\text{g L}^{-1}$) for each catchment. Blue points indicate years with outbreaks while red represents no outbreaks.

Figure 4: This is great visualization of the magnitude of herbivory effect. n increased and decreased adds up only to 262 – so was there no change at all for 4 lakes? It is not very clear to me how did you determine the absolute change in DOC, maybe it could be clarified in methods.

We thank the Reviewer for catching this missing detail in the figure caption. We have adjusted the location of the axis ticks to make it clearer that zero is not included in the figure.

Added line 218

“Figure 4. Defoliator outbreaks offset increases in lake water DOC concentrations observed across the northern hemisphere. Bars represent number of lakes that showed either an increase (black; n = 247) or decrease (open; n = 15) in DOC between 1990 and 2016⁴⁴. **Four lakes showed no change in DOC and are omitted from the plot.** Vertical dashed line is the mean decline in DOC averaged across ice-free months estimated during defoliator outbreaks over the same study period from 12 catchments in Ontario, Canada. Shaded area represents the 95% CI for the change in DOC calculated via bootstrapping.”

We have also included more detail in the Methods on how the absolute change in DOC and DIN were calculated.

Added lines 456-466

“For each lake, we calculated the absolute Theil-Sen slope for the inter-annual trend in DOC concentrations between 1990 and 2016. This approach provides a non-parametric estimate of the slope of a linear regression line fitted through two variables that is more robust to outliers. We compared these values to the change (i.e. slope) in mean annual DOC concentrations associated with complete catchment disturbance. The effect of insect damage was estimated from mixed effects models fitted to our 12 study lakes between 1990 and 2016 (see below for fitting procedure). For DIN, we calculated between-year trends from our 12 study lakes because the wider ICP waters dataset does not sample DIN. Like for DOC, we calculated the absolute Theil-Sen slope for the temporal trend in mean annual DIN in each lake and compared it to the estimated within-year effect of insect defoliation.”

Figure 5: Can you see any trends for increased frequency of outbreaks from this data (or from complete annual herbivory mapping data collected since 40s)?

We see no trend in outbreak frequency so have not mentioned this in the text.

Line 312-315, 323-325: Are there any studies on why some catchments are, or would be in the future, more frequently disturbed?

Temperature and precipitation are predicted to have the greatest impact on insect outbreaks, as well as forest disturbance in general (Seidl et al. 2017 *Nat Clim Chang*). Since climate change will continue to shift the temperature and precipitation patterns across the boreal region, it likely that catchments may experience different climatic patterns that may increase or decrease the frequency and severity of disturbances.

Added lines 330-340

“Second, the large range we found in the frequency of insect outbreaks may promote heterogeneity in catchment biogeochemistry across the landscape⁷². Defoliation leads to a loss of nutrients from catchments by transferring OM from forests to lakes³⁷, ultimately

shifting the nutrient balance of C and N in lakes. More frequently defoliated catchments may therefore experience a phase shift in lake biogeochemistry relative to catchments with less frequent outbreaks. This divergence stands to be enhanced by climate change. Forests in the study region also face an increased risk of defoliator outbreaks⁷³ due to the range expansion and increased population growth of insects associated with climate change⁷⁴. Warmer temperatures may also disrupt the relationship between trees and insects (e.g., phenology, plant defence) which will shift insects to areas more favourable to outbreaks⁷⁴.”

Methods/insect damage: It is told here that outbreaks were aerially mappable when moderate to severe defoliation was observed, defined by the absence of at least 25% of foliar cover, and that extent and severity of insect activity was mapped. Thus, it seems that some outbreak areas have been classified as more severely defoliated than others but did you take the severity of defoliation into account when the proportion of disturbance was calculated? The Reviewer is correct that we did not include severity of disturbance (i.e. the categorical measure of foliage lost as determined by aerial observation). We did not include it in our model because 90% of all the outbreak events in our study were classified as “moderate-severe”. Due to the lack of variation in severity measures, we focused on the proportion of the catchment disturbed as our measure of intensity.

Added lines 379-383

“As 90% of disturbance events were classified as moderate-severe in our dataset, there was effectively no variation in damage severity. We therefore calculated the proportion of each study catchment that was damaged by insects to quantify insect disturbance in subsequent analyses.”

Supplementary Table 1: Treed should probably be trees, right?

Changed as suggested

Supplementary Figure 1, line 731: should it be from 2004 to 2004?

Yes, there was a typo here that we have now corrected. The years should read 2004 to 2016. Thank you for catching this mistake.

Change made on lines 763-765

“A paired t-test was used to compare the mean difference in the annual area covered between the two groups from 2004 to **2016** ($t_{14} = - 8.70$, $p < 0.001$).”

Reviewer #3 (Remarks to the Author):

The manuscript “Forest defoliator outbreaks disrupt nutrient cycling in northern waters” by Woodman *et al.* presents analysis of long-term trends in outbreaks of defoliating insects and inputs to northern watersheds. The authors present analysis of data from more than 10,000 watersheds over a period of nearly 30 years. The authors found that the type of tree (gymnosperms vs. angiosperms) affected the amount of dissolved carbon released by defoliators.

Overall, the manuscript is well-written and with climate change and increasing frequency of gross-tissue removing (defoliating) insects, the results of this study will be valuable to a wide readership and will provide an excellent baseline for follow-up studies.

We thank the Reviewer for appreciating the breadth and novelty of our work.

Major concerns:

Although I understand the authors use of colors to represent forest colors, the figure colors need to be updated and different symbols used (Figure 2) because the barely different colors do not stand out.

We have increased the thickness of the lines and changed the point shapes and color to allow for easier interpretation. These changes have been applied to Figures 1, 2, and Supplementary Figure 2 since they all use the same aesthetic. Only Figure 1 is shown here to save space.

Original figure on left, revised figure on right
Some information should be added about the types of lakes. Are these lakes deep and thermally stratified? Is there constant uptake by algae. Introduction should include more about phosphorus nitrogen ratios and algal response.

The information about lake type requested by the Reviewer have been added to the “Study Sites” section of the Methods rather than the Introduction. We believe grouping these new details with existing information in the Methods will make finding and interpreting information about our study lakes easier.

Added lines 351-352

“All our study lakes thermally stratify except for Wishart Lake.”

Changed lines 352-354

“Lake waters were nutrient-poor, with total phosphorus concentrations between 3.1 to 6.0 $\mu\text{g L}^{-1}$ and N:P ratios between 32 to 104 measured via monthly surface grabs at the point of maximum depth.”

Change to Supplementary Table 3

Supplementary Table 3. Summary of study lakes. Asterisk (*) **denotes lake that is not thermally stratified.**

Region	Lake	Latitude (°)	Longitude (°)	Lake Area (ha)	Max depth (m)	Catchment Area (ha)
	Little					
Algoma	Turkey	47.04227	-84.40812	19.85	13.0	134
Algoma	Wishart*	47.04948	-84.39924	19.09	4.5	245
Greater Sudbury	Clearwater	46.37042	-81.05045	77.02	21.5	285
Greater Sudbury	Hannah	46.44328	-81.03831	27.80	8.0	83
Greater Sudbury	Lohi	46.38749	-81.04330	41.15	19.0	83
Greater Sudbury	Middle	46.43897	-81.02482	28.98	12.0	132
Greater Sudbury	Sans					
Greater Sudbury	Chambre	46.72141	-81.13066	15.72	17.5	62
Greater Sudbury	Swan	46.36602	-81.06544	7.48	8.8	18
Muskoka	Blue Chalk	45.19917	-78.93835	49.55	22.0	1045
Muskoka	Crosson	45.08356	-79.03645	54.96	26.0	508
Muskoka	Red Chalk	45.18981	-78.94753	55.10	32.0	381
Temiskaming	Whitepine	47.38400	-80.63133	87.24	59.0	911

Similarly, we agree with the Reviewer's suggestion to include more detail on nutrient ratios and algal uptake. However, we believe the latter is more relevant to the Discussion than the Introduction, because we do not analyse algal responses in the paper, i.e. they are an implication of our Results, not pertinent background to interpret the Results.

Added lines 65-69

“Unlike C and N, there may be little change in phosphorus (P) cycling from defoliation as it relies more on atmospheric inputs, wetland flowpaths, and internal recycling than plant litter fluxes²⁸. N:P ratios may therefore shift downstream, with consequences for lake food webs as P limitation becomes more prevalent²⁹.”

Added lines 307-309

“Lakes in our study region remain stratified until late-autumn⁶⁴, during which algae continue to uptake nutrients^{65,66} and respond to changes in overlying water quality associated with DOC inputs.”

Changes lines 309-312

“During severe insect outbreaks, **algal** productivity in lakes may **therefore** increase **in late-autumn** due to elevated transfer of N from forests via frass and less light attenuation by DOC from reduced litterfall inputs.”

Lines 300-303. Epilemmetic DOC suggests lakes (at least some) are thermally stratified. The authors need to address language in the discussion as their thoughts are unclear. Are the results observed in water quality similar across lakes or a source of variation. The timing of defoliation should be a period during stratification and leaf fall should occur as the stratification breaks down.

We agree that the language here was unclear. We have removed all mention of “epilimnion” in the paper and better explained in the Discussion how the timing of stratification and leaf fall interact with the potential for algal uptake of nutrients. The full changes on lines 309-312 are given immediately above in response to the previous comment.

We have also clarified that our mixed effects modelling approach accounts for the variation across lakes, such as in the timing of stratification events, and therefore the observed results capture the overall effect of insect outbreaks on lake biogeochemistry.

Added to lines 479-483

“We accounted for variation among sites, such as associated with differences in disturbance history, climate, timing of stratification events, and soil type, by including catchment identity as a random effect. Including catchments as a random effect ensures that our model output represents the overall effect of insect outbreaks on lake biogeochemistry while incorporating the variation across catchments.”

It is surprising that the authors did not observe changes in other nutrients associated with outbreaks of defoliators. Phosphorus is a key nutrient which appeared unaffected. Why would this be the case?

Added lines 65-67

“Unlike C and N, there may be little change in phosphorus (P) cycling from defoliation as it relies more on atmospheric inputs, wetland flowpaths, and internal recycling than plant litter fluxes²⁸.”

Added lines 270-273

“The lack of an effect of insect outbreaks on lake P concentrations is likely due to atmospheric deposition and wetland export being the primary sources of P in our system rather than allochthonous inputs from forests²⁸.”

Although the authors cite a large number of papers, I feel that a number of important papers are not cited. Perhaps they could pair down their current references and add some of the other papers here.

As recommended, we removed 14 references in total, particularly from the Introduction. We added 13 references to our updated Discussion to address Reviewer suggestions, including all those suggested below by Reviewer 3.

With climate change, plants are producing more foliage but in natural conditions, the foliage is often nutrient poor. The authors should reference changes in plant nutrition associated with elevation in CO₂. There is a growing body of literature but the paper by Newall *et al.* 2018 *Aquatic Sciences* 80: 27 and references within would help with interpretations.

We have included details on how climate change may lower leaf nutrient content which, in turn, may weaken the impact of insect outbreaks on lake biogeochemistry. We thank the reviewer for this insight.

Added lines 288-289

“However, as elevated CO₂ levels reduce leaf nutrient content⁵⁶, frass produced from insect outbreaks may become less nutrient-rich ”

In addition, Lapierre *et al.* 2013 *Nature Communications* is very relevant to interpreting changes in aquatic DOC that may also be influenced by atmospheric CO₂.

We have also included details on how elevated CO₂ will increase export of terrestrial OM to aquatic systems by increasing forest cover and runoff from soils using Newall *et al.* 2018 and references within Lapierre *et al.* 2013. We thank the reviewer for their recommendations.

Added lines 318-322

“Ongoing climate change and rising CO₂ concentrations will further increase terrestrial OM export to lakes by promoting soil runoff⁶⁸ and forest growth⁶⁹, respectively. These increases in terrestrial OM and the associated accumulation of DOC⁷⁰ should also promote heterotrophic growth⁶⁸ in the absence of insects.”

Recently, DOC (a main component measured by the authors) has been linked to browning regardless of other factors. See Kritzberg *et al.* 2020. *Ambio* 49: 375-390. The authors should consider if increasing DOC in northern climates is affecting water clarity regardless of bacteria.

We agree that it is important to consider the effects of increasing DOC concentrations linked to browning. We have addressed future “accumulation” of DOC independent of “insects”, which we think the Reviewer meant instead of “bacteria”.

Added lines 318-322

“Ongoing climate change and rising CO₂ concentrations will further increase terrestrial OM export to lakes by promoting soil runoff⁶⁸ and forest growth⁶⁹, respectively. These increases in terrestrial OM and the associated accumulation of DOC⁷⁰ should also promote heterotrophic growth⁶⁸ in the absence of insects.”

Added lines 326-328

“First, the reduction in litterfall inputs to lakes from insects should provide a within-year pause in DOC accumulation that enables higher whole-lake productivity as more light and nutrients are available for algal uptake.”

The authors need to better connect the results of the study to the broad implications of accumulating DOC (lines 308-309) and being paused by defoliator outbreaks. In the context of overall lake health, are defoliators good?

We have better connected the results of the study to the broad implications of accumulating DOC. We think that this is more useful than prescribing a “good” or “bad” connotation to defoliator impacts, which we think are a much more complex idea. For example, what may seem like a positive for the lake food web (e.g. increased productivity from more light and nutrients) may lead to greater CO₂ emissions, which is less “healthy” for the broader landscape. Rather, we believe our results highlight the importance of insect outbreaks in altering within-year nutrient cycling and for providing a source of variation in lake biogeochemistry across the landscape, particularly in the context of browning.

Added lines 326-328

“First, the reduction in litterfall inputs to lakes from insects should provide a within-year pause in DOC accumulation that enables higher whole-lake productivity as more light and nutrients are available for algal uptake.”

Added lines 330-335

“Second, the large range we found in the frequency of insect outbreaks may promote heterogeneity in catchment biogeochemistry across the landscape⁷². Defoliation leads to a loss of nutrients from catchments by transferring OM from forests to lakes³⁷, ultimately shifting the nutrient balance of C and N in lakes. More frequently defoliated catchments may therefore experience a phase shift in lake biogeochemistry relative to catchments with less frequent outbreaks.”

Minor comments:

Line 34: macronutrients, including carbon – Changed as suggested

Line 39: by affecting (not controlling) – Changed as suggested

Line 75: sites because – Changed as suggested

Line 168: - a 45% (typo?) – Yes, this was a typo based on an outdated analysis and we have updated the text. The true values is 32%. It is the difference between the DOC at 0 and 100% damage for 60% deciduous forest cover seen from the black line in Fig. 3. The corresponding DOC concentrations are 3.55 and 2.43 mg L⁻¹, respectively. Thank you for catching this mistake.

Changed on lines 174-176

At the upper quantile of deciduous tree cover (=0.60), this reduction reached 1.13 (95% CI: 0.28 to 2.00) mg L⁻¹ over the range of insect damage – a 32% reduction on a relative basis (i.e. black line, Figure 3).

Line 238: nutrient cycling in temperate zones – We have avoided specifying a specific region so as not to give the impression that we are over-generalizing across all temperate forests, which Reviewer 1 was concerned about.

Line 248: nutrient rich frass has some important nutrients, nitrogen not phosphorus for example – We have removed this part of the sentence as it repeated the first sentence of the paragraph, so this comment is no longer relevant.

Line 258: even small additions of N will result in proportionally large increases in concentrations that will be biologically important. This is unclear. Concentrations of available Nitrogen? Biomagnification? This seems important but very unclear.

Change on lines 268-270

“Given that northern lakes are often N-limited⁵³, even small additions of N will result in proportionally large increases in available N concentrations that can promote the productivity of aquatic food webs²⁹.”

Line 282: mechanisms by which they achieve that differ. Unclear what is meant.

Change on lines 295-297

“While defoliating and wood boring insects can both increase N concentrations in downstream waters^{12–15}, **these increases arise through different mechanisms between the two insect types.**”

Line 301: The connection of DOC to decreased bacteria favoring pelagic algal species needs to be better explained or cut from discussion. – **Cut as suggested.**

W. Wyatt Hoback

REVIEWERS' COMMENTS

Reviewer #1 (Remarks to the Author):

As I wrote in the previous peer review, I appreciate the fact that they are able to robustly extract the effects of defoliation by 32 years of long-term data for 12 lakes in Ontario, which they discuss in their data, and I believe that this finding is worthy of publication.

I think the two major issues I pointed out in the previous review have been addressed in the text of the new manuscript.

However, there are still two points that concern me.

1. Are the "266 lakes in the Northern Hemisphere" selected from the ICP water report geographically located in areas that can be called "northern waters"? I think "Northern Hemisphere" generally refers to a larger area than the area where Ontario is located. This implies that those groups of lakes have a much higher diversity of climatic environments.

2. In the abstract section of the revised manuscript, the description of the conclusions from this extensive data analysis has changed significantly from the first draft; this is not a change in the level of expression, but a change in the logic of the argument to be made.

If the reviewer's point is correct, it is important to respond to it, but changes to the conclusion are generally not acceptable.

Reviewer #2 (Remarks to the Author):

Thank you for your thorough job of addressing the concerns and remarks of the reviewers. This revision has addressed all my previous comments. I think that the manuscript is now even clearer. Still happened to spot one very minor thing in line 409, shouldn't it read? "First, the reduction in litterfall inputs to lakes due to insects should provide..."

Reviewer #3 (Remarks to the Author):

The revisions by Woodman et al. improve and clarify the manuscript. I carefully reviewed all changes and responses to the 3 reviews and I am satisfied that this contribution is scientifically rigorous and of general interest to readers of Nature Communications.

Reviewer #1 (Remarks to the Author):

As I wrote in the previous peer review, I appreciate the fact that they are able to robustly extract the effects of defoliation by 32 years of long-term data for 12 lakes in Ontario, which they discuss in their data, and I believe that this finding is worthy of publication.

I think the two major issues I pointed out in the previous review have been addressed in the text of the new manuscript.

We again thank the Reviewer for appreciating the merit of our work and improving the previous draft.

However, there are still two points that concern me.

1. Are the "266 lakes in the Northern Hemisphere" selected from the ICP water report geographically located in areas that can be called "northern waters"? I think "Northern Hemisphere" generally refers to a larger area than the area where Ontario is located. This implies that those groups of lakes have a much higher diversity of climatic environments. We agree that think "Northern Hemisphere" generally refers to a larger area than the area where Ontario is located and thank the Reviewer for pointing this out. The 266 lakes are all from areas that can be called "northern waters". We have replaced all instances of "northern hemisphere" with "boreal and north temperate" to make this point explicit. See lines 26, 245, 249, 533, and 889 for location of change.

2. In the abstract section of the revised manuscript, the description of the conclusions from this extensive data analysis has changed significantly from the first draft; this is not a change in the level of expression, but a change in the logic of the argument to be made. The logic of our arguments has not changed between revisions (bolding shows change).

Initial submission:

"As insect defoliator outbreaks occur at least every 5 years across a wider 439,661 km² boreal ecozone, we suggest they are an underappreciated driver of biogeochemical cycles in forest catchments."

Revised submission:

"As insect defoliator outbreaks occur at least every 5 years across a wider 439,661 km² boreal ecozone **of Ontario**, we suggest they are an underappreciated driver of biogeochemical cycles in forest catchments **of this region**."

Rather, we contextualized our results by adding "**of Ontario**" and "**of this region**", as Reviewer 1 originally requested. Therefore, both our logic and key result – that insect outbreaks are an underappreciated driver of biogeochemical cycling – remain unchanged.

Reviewer #2 (Remarks to the Author):

Thank you for your thorough job of addressing the concerns and remarks of the reviewers. This revision has addressed all my previous comments. I think that the manuscript is now even clearer. Still happened to spot one very minor thing in line 409, shouldn't it read? "First, the reduction in litterfall inputs to lakes due to insects should provide...."

We thank the Reviewer for their comments and have made the suggested change on line 405.

Reviewer #3 (Remarks to the Author):

The revisions by Woodman *et al.* improve and clarify the manuscript. I carefully reviewed all changes and responses to the 3 reviews and I am satisfied that this contribution is scientifically rigorous and of general interest to readers of Nature Communications.

We thank the Reviewer for their comments which have helped improve the manuscript.